

# Advertisement call of *Brachycephalus albolineatus* (Anura: Brachycephalidae)

Marcos R. Bornschein[1,2], Luiz Fernando Ribeiro[2,3], Mario M. Rollo Jr[1], André E. Confetti[4] and Marcio R. Pie[2,5]

[1] Instituto de Biociências, Universidade Estadual Paulista, São Vicente, São Paulo, Brazil
[2] Mater Natura - Instituto de Estudos Ambientais, Curitiba, Paraná, Brazil
[3] Pontifícia Universidade Católica do Paraná, Escola de Ciências da Vida, Curitiba, Paraná, Brazil
[4] Universidade Federal do Paraná, Programa de Pós-Graduação em Zoologia, Curitiba, Paraná, Brazil
[5] Universidade Federal do Paraná, Departamento de Zoologia, Curitiba, Paraná, Brazil

Corresponding author
Marcos R. Bornschein,
bornschein.marcao@gmail.com

## ABSTRACT

**Background:** *Brachycephalus* are among the smallest terrestrial vertebrates in the world. The genus encompasses 34 species endemic to the Brazilian Atlantic Rainforest, occurring mostly in montane forests, with many species showing microendemic distributions to single mountaintops. It includes diurnal species living in the leaf litter and calling during the day, mainly during the warmer months of the year. The natural history of the vast majority of the species is unknown, such as their advertisement call, which has been described only for seven species of the genus. In the present study, we describe the advertisement call of *Brachycephalus albolineatus*, a recently described microendemic species from Santa Catarina, southern Brazil.

**Methods:** We analyzed 34 advertisement calls from 20 individuals of *B. albolineatus*, recorded between 5 and 6 February 2016 in the type locality of the species, Morro Boa Vista, on the border between the municipalities of Jaraguá do Sul and Massaranduba, Santa Catarina, southern Brazil. We collected five individuals as vouchers (they are from the type series of the species). We used the note-centered approach to describe the advertisement calls of the species.

**Results:** *B. albolineatus* have a long advertisement call of 40–191 s (mean of 88 s) composed of 8–29 notes (mean of 17 notes) emitted at a rate of 6–18 notes per minute (mean of 11 notes per minute) and at a note dominant frequency of five to seven kHz (mean of six kHz). Advertisement calls are composed of isolated notes and note groups (two notes involved in each particular note group); the former is composed by one to three pulses (mean of 2.0) and the note groups by two or three pulses in each note (mean of 2.7). Most advertisement calls present both isolated notes and note groups, with a few cases showing only the former. Note groups are emitted invariably in the last third of the advertisement call. Most isolated notes escalate their number of pulses along the advertisement call (1–2, 1–3 or 2–3). Note duration of isolated notes varies from 0.002 to 0.037 s (mean of 0.020 s) and duration of note group vary from 0.360 to 0.578 s (mean of 0.465 s).

**Discussion:** Individuals increase the complexity of their calls as they proceed, incorporating note groups and pulses per note. Intra-individual variation analysis also demonstrated that less structured advertisement calls (i.e., with notes with fewer pulses) are not stereotyped. It is possible that isolated notes and note groups could

have distinct functions, perhaps territorial defense and mating, respectively. We argue that using a note-centered approach facilitates comparisons with calls of congeners, as well as underscores the considerable differences in call structure between species in a single group and among species groups.

## INTRODUCTION

*Brachycephalus* are among the smallest terrestrial vertebrates in the world (*Rittmeyer et al., 2012*), with most species not exceeding 2.5 cm in body length. The genus includes 34 species (*Frost, 2017*), occurring from the southern Bahia to northeastern Santa Catarina, Brazil (*Bornschein et al., 2016a*; *Pie et al., 2013*). Most *Brachycephalus* species, particularly in the *Brachycephalus pernix* species group (see below), are microendemic, occurring in one or a few adjacent mountaintops, with total extents of occurrence comparable to the smallest ranges of species around world (*Bornschein et al., 2016a*). Species are diurnal, living in the leaf litter in forests of the Atlantic Rainforest domain (*Bornschein et al., 2016a* and compilation therein). Direct development, with a reduced number of eggs laid on the soil (*Pombal, 1999*), was demonstrated for *B. ephippium* (*Heyer et al., 1990*; *Pombal, 1999*), and this is assumed as the reproductive pattern for the genus. *Brachycephalus* is characterized by extreme miniaturization, which is possible related to a reduced number and size of digits (*Hanken & Wake, 1993*; *Yeh, 2002*; *Clemente-Carvalho et al., 2009*) and loss of some morphological features of the auditory apparatus (*Silva, Campos & Sebben, 2007*). Some species are brightly colored, with neurotoxins found in the skin of two aposematic species (*Sebben et al., 1986*; *Pires et al., 2002*, *2003*, *2005*; *Schwartz et al., 2007*), possibly originated from intestinal bacteria (*Schwartz et al., 2007*). The species of the genus have been segregated into three phenetic groups, namely the *B. ephippium*, *B. didactylus*, and *B. pernix* species groups (*Ribeiro et al., 2015*). Possibly due to historical evolutionary processes (*Bornschein et al., 2016a*; *Firkowski et al., 2016*), *Brachycephalus* species are almost exclusively allopatric or parapatric, with few cases of syntopy (*Bornschein et al., 2016a*).

There has been a recent increase in the description of new species within *Brachycephalus*, with 20 species described in the last 10 years (*Frost, 2017*). However, the natural history of the vast majority of the species is unknown (see review of ecological studies in *Bornschein et al., 2016a*). Call descriptions of the species are scarce, which is surprising, given that individuals of the species are usually located by their calls, often emitted at locally high male densities (one person might hear dozens of males from a single hearing spot). Advertisement calls were described for *Brachycephalus ephippium* (*Pombal, Sazima & Haddad, 1994*; *Goutte et al., 2017*), *B. hermogenesi* (*Verdade et al., 2008*), *B. pitanga* (*De Araújo et al., 2012*; *Tandel et al., 2014*; *Goutte et al., 2017*), *B. tridactylus* (*Garey et al., 2012*), *B. crispus* (*Condez et al., 2014*), *B. sulfuratus* (*Condez et al., 2016*), and *B. darkside* (*Guimarães et al., 2017*).

Given that *Brachycephalus* is a group with mostly allopatric species, it is of great interest to investigate the evolution pattern of their calls. In allopatry, one could expect great similarity between the call of different species (*Bornschein et al., 2007*; *Maurício et al., 2014*), due to a lack of selective pressure to avoid hybridization of closely-related species. However, this needs to be tested for *Brachycephalus*. In the present study, we describe the advertisement call of *Brachycephalus albolineatus*, a member of the *B. pernix* group (*Bornschein et al., 2016b*). *B. albolineatus* was recently described based on a series of eight specimens collected at the type locality, Morro Boa Vista, Santa Catarina, southern Brazil (*Bornschein et al., 2016b*).

## METHODS

We recorded individuals of *B. albolineatus* on 25 October 2012 and on 5–6 February 2016 at the type locality of the species, i.e., Morro Boa Vista (26°30′58″S, 49°03′14″W; 820–835 m above sea level), on the border between the municipalities of Jaraguá do Sul and Massaranduba, state of Santa Catarina, southern Brazil. We collected vouchers according to permits issued by ICMBIO–SISBIO (no. 20416–2). Vouchers belong to the type material of the species, which was deposited in Museu de História Natural Capão da Imbuia (MHNCI), Curitiba, Paraná state and Museu Nacional (MNRJ), Rio de Janeiro, Rio de Janeiro state, Brazil. Analyzed recordings were carried out on 5–6 February 2016 from 9:00 to 12:00 a.m. and from 15:00 to 18:00 p.m. Climatic conditions during recordings were characterized by air temperature = 20.8–21.4 °C, soil temperature = 19.4–20.0 °C, and relative air humidity = 86–100%. We made numbered markings on the vegetation above the recorded individuals in the field to determine whether new recordings were from the same individuals to build up the dataset both in terms of more individuals as well as intra-individual variation, with more than one recording from the same individual. Calls were recorded using the digital recorders Sony PCM-D50 and PCM-M10, both with sampling frequency rate of 44.1 kHz and 16-bit resolution, and Sennheiser ME 66 microphones. Recordings were deposited in MHNCI. Sound samples were analyzed with Raven Pro 1.5 (*Bioacoustics Research Program, 2012*). Time domain variables were measured from oscillograms and frequency domain variables were measured from spectrograms. Spectrogram features were defined with a 256-point Fast Fourier Transform (FFT), a 3-dB Filter bandwidth of 492 Hz, Hann window, 50% overlap, and a spectrogram color scheme of Standard Gamma II. Final spectrograms, as well as diagnostic plots, were generated using the Seewave package, v. 2.0.5 (*Sueur, Aubin & Simonis, 2008*) of the R environment, v. 3.2.2 (*R Core Team, 2015*) using the same window size and overlap settings as in Raven Pro, but resampling the audio files at 22.05 kHz.

We used the note-centered approach *sensu Köhler et al. (2017)* to define the advertisement call of the species. We determined the end of a given call and the beginning of the next one by the long period of silence between them (*Köhler et al., 2017*), which might last for several minutes and thus is considerably longer than the call itself. We described the advertisement calls following features and criteria of *Köhler et al. (2017)*. We took the liberty of describing the general features of *Köhler et al. (2017)* also for parts of the call, in order to clarify the distinctions observed in particular parts of the advertisement

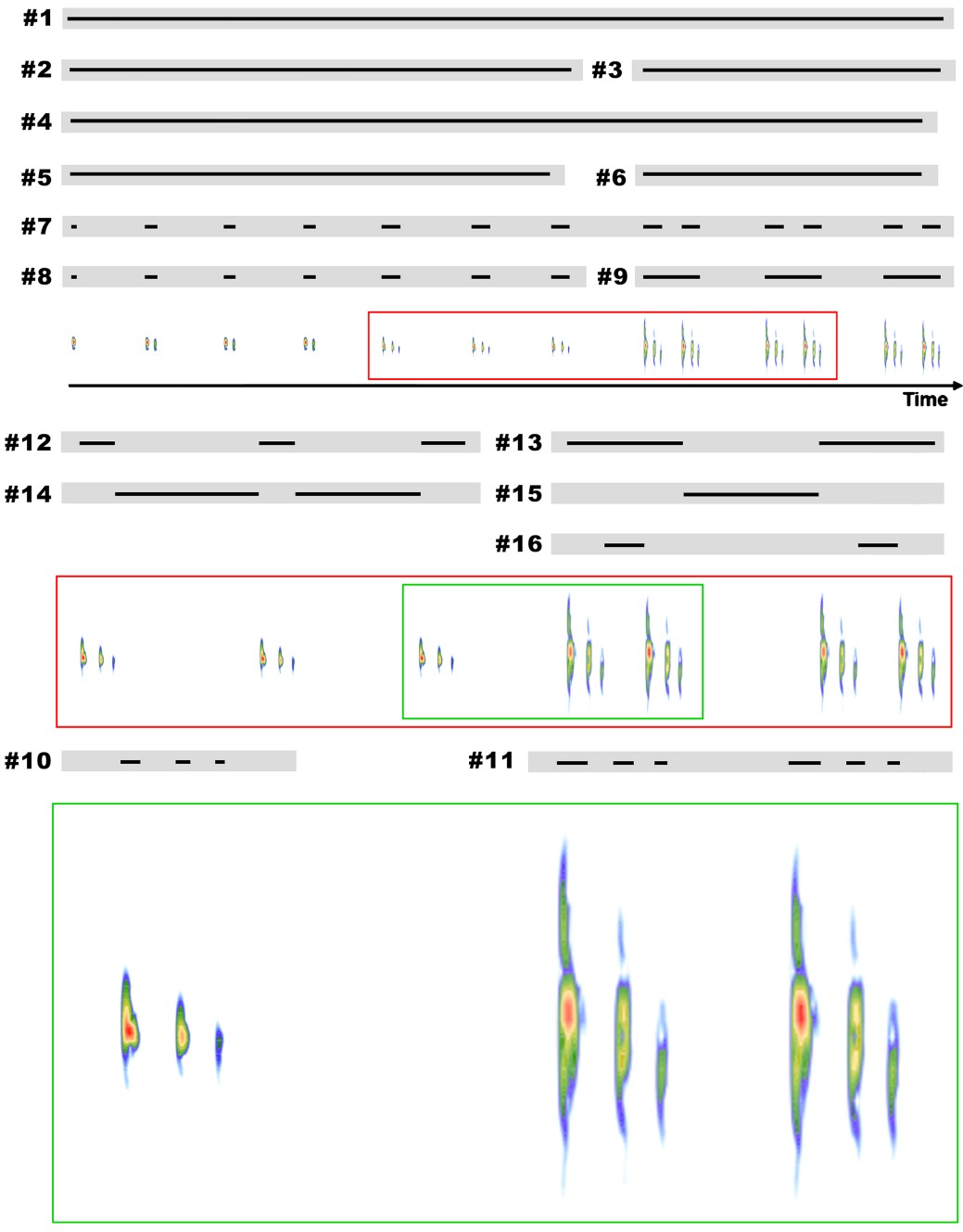

**Figure 1 Representation of some features considered in the description of the advertisement call of *B. albolineatus* on a schematic call.** Numbers correspond to the order of descriptions in the methods. (1) Call duration (s); (2) duration of the call including only isolated notes (s); (3) duration of the call including only note groups (s); (4) note rate (notes per minute); (5) note rate of the call including only isolated notes (notes per minute); (6) note rate of the call including only note groups (notes per minute); (7) number of notes per call (10 notes in the example); (8) number of isolated notes per call (seven notes in the example); (9) number of note groups per call (three notes in the example); (10) number of pulses per isolated notes (three in the example); (11) number of pulses in each note groups (3–3 in the example); (12) note duration of isolated notes (s); (13) duration of note group (s); (14) inter-note interval in isolated notes (s); (15) inter-note group interval (s); and (16) inter-note interval within note groups (s).

calls of *B. albolineatus*. We used the following features, which can be seen in Fig. 1: (1) call duration (s); (2) duration of the call including only isolated notes (s); (3) duration of the call including only note groups (s); (4) note rate (notes per minute); (5) note rate of the call including only isolated notes (notes per minute); (6) note rate of the call including only note groups (notes per minute); (7) number of notes per call; (8) number of isolated notes per call; (9) number of note groups per call; (10) number of pulses per isolated notes; (11) number of pulses per note in note groups; (12) note duration of isolated notes (s); (13) duration of note group (s); (14) inter-note interval in isolated notes (s), defined as the time from the end of one isolated note to the beginning of the next isolated note; (15) inter-note group interval (s), defined as the time from the end of one note group to the beginning of the next note group; (16) inter-note interval within note groups (s), defined as the time from the end of the first note to the beginning of the next note of the same note group; (17) note dominant frequency (kHz); (18) highest frequency (kHz) and (19) lowest frequency (kHz). The note rate was calculated taking into account the time from the beginning of the first note to the beginning of the last note of the calls (or call intervals) and the number of notes included in this counted time (the last note is not included; *Köhler et al., 2017*; *Cocroft & Ryan, 1995*). The dominant frequency across all notes in a call sample was calculated with the function dfreq from the R package seewave. This function brings as an output a plot with all dominant frequencies in a specific file or file segment. Alternatively, the output can be a vector of dominant frequency values. All the default arguments of the function were followed, with the exception of the overlap, for which we chose the value of 90% and the amplitude threshold of signal detection, whose value we determined as of 5%. We measured the highest and lowest frequencies from notes.

## RESULTS

We recorded calls from 29 individuals of *B. albolineatus* but analyzed 34 advertisement calls from 20 individuals, five of which were collected as vouchers (MHNCI 10296–9, MNRJ 90349). We recorded eight individuals two to four times ($\bar{x} = 2.75$ times per individual). The calls we deposited resulted in 34 separate recordings (MHNCI 001–34).

*B. albolineatus* emitted a relatively long advertisement call, between 39.93 and 191.14 s ($\bar{x} = 88.37 \pm 35.73$ s; Table 1; see feature #1 in Fig. 1). Thereafter, the individual remains silent for several minutes, occasionally for more than 35 min, when it emits a new advertisement call. A graphical representation of the temporal sequence of notes in each call is shown in Fig. 2. The note rate was 5.89–18.09 notes per minute ($\bar{x} = 11.44 \pm 3.22$ notes per minute; Table 1; see feature #4 in Fig. 1). Advertisement calls included 8–29 notes ($\bar{x} = 17.26 \pm 6.38$ notes; Table 1; see feature #7 in Fig. 1).

The advertisement calls of the species included both isolated notes and note groups (in this case, with two notes involved in each particular note group; Fig. 3). Advertisement calls could be composed only by isolated notes (23.5% of advertisement calls), but usually included both isolated notes and note groups (Table 2). Every advertisement call with isolated notes and note groups began with

**Table 1 Measurements of advertisement call (AC) features of *B. albolineatus* and some parameters.**

| Feature/[Analysis] | Range | Mean | SD | N Samples | Individuals |
|---|---|---|---|---|---|
| Call duration (s) (1) (entire call) | 39.933–191.141 | 88.367 | 35.733 | 24 | 16 |
| Duration of the call including only isolated notes (s) (2) when note groups is absent | 49.971–191.141 | 100.675 | 52.423 | 6 | 6 |
| Duration of the call including only isolated notes (s) (2) when note groups occurs | 18.387–98.896 | 53.709 | 25.380 | 18 | 13 |
| Duration of the call including only note groups (s) (3) | 0.407–60.317 | 23.416 | 17.867 | 25 | 16 |
| Note rate (notes per minute) (4) (entire call) | 5.891–18.088 | 11.439 | 3.216 | 24 | 16 |
| Note rate of the call including only isolated notes (notes per minute) (5) when note groups is absent | 5.891–9.879 | 7.707 | 1.707 | 6 | 6 |
| Note rate of the call including only isolated notes (notes per minute) (5) when note groups occurs | 7.282–13.619 | 10.254 | 1.611 | 18 | 13 |
| Note rate of the call including only note groups (notes per minute) (6) | 12.810–27.162 | 18.603 | 3.864 | 20 | 14 |
| Number of notes per call (7) | 8–29 | 17.26 | 6.38 | 27 | 16 |
| Number of isolated notes per call (8) | 4–26 | 11.04 | 4.57 | 27 | 16 |
| Number of note groups per call (9) | 0–9 | 3.38 | 2.94 | 26 | 16 |
| (Percentage of number of notes of the entire AC included in note groups in each AC) | 0.00–76.19 | 31.67 | 24.36 | 26 | 16 |
| Number of pulses per isolated notes (10) | 1–3 | 2.00 | 0.595 | 323 | 20 |
| (Number of isolated notes with one pulse) | 26 | – | – | 324 | 20 |
| (Number of isolated notes with two pulses) | 187 | – | – | 324 | 20 |
| (Number of isolated notes with three pulses) | 110 | – | – | 324 | 20 |
| Number of pulses per note in note groups (11) | 2–3 | 2.70 | 0.459 | 230 | 16 |
| (Number of notes of note groups with 2–2 pulses) | 25 | – | – | 115 | 16 |
| (Number of notes of note groups with 2–3 pulses) | 5 | – | – | 115 | 16 |
| (Number of notes of note groups with 3–3 pulses) | 71 | – | – | 115 | 16 |
| (Number of notes of note groups with 3–2 pulses) | 14 | – | – | 115 | 16 |
| (Total number of pulses in note groups) | 4–6 | 5.40 | 0.825 | 115 | 16 |
| Note duration of isolated notes (s) (12) | 0.002–0.037 | 0.020 | 0.007 | 96 | 19 |
| Duration of note groups (s) (13) | 0.360–0.578 | 0.465 | 0.053 | 62 | 16 |
| Inter-note interval in isolated notes (s) (14) | 4.092–12.248 | 6.663 | 1.705 | 62 | 15 |
| Inter-note group interval (s) (15) | 4.322–10.678 | 6.871 | 1.768 | 32 | 13 |
| Inter-note interval within note groups (s) (16) | 0.319–0.526 | 0.412 | 0.050 | 55 | 16 |
| Note dominant frequency (kHz) | 5.340–7.321 | 6.376 | 0.304 | 256 | 10 |
| Highest frequency (kHz) | 7.113–9.852 | 8.437 | 0.492 | 145 | 19 |
| Lowest frequency (kHz) | 3.092–5.212 | 4.066 | 0.448 | 145 | 19 |

**Note:**
The number between brackets represent the number of the feature in Fig. 1. SD, standard deviation.

the former and then changed to note groups (Table 2; Fig. 2). Note groups contains, on average, 31.7% of the notes of the entire advertisement call (±24.4%; range of 0–76.2%; Table 1; see feature #9 in Fig. 1) and span, on average, 23.42 s (±17.87 s;

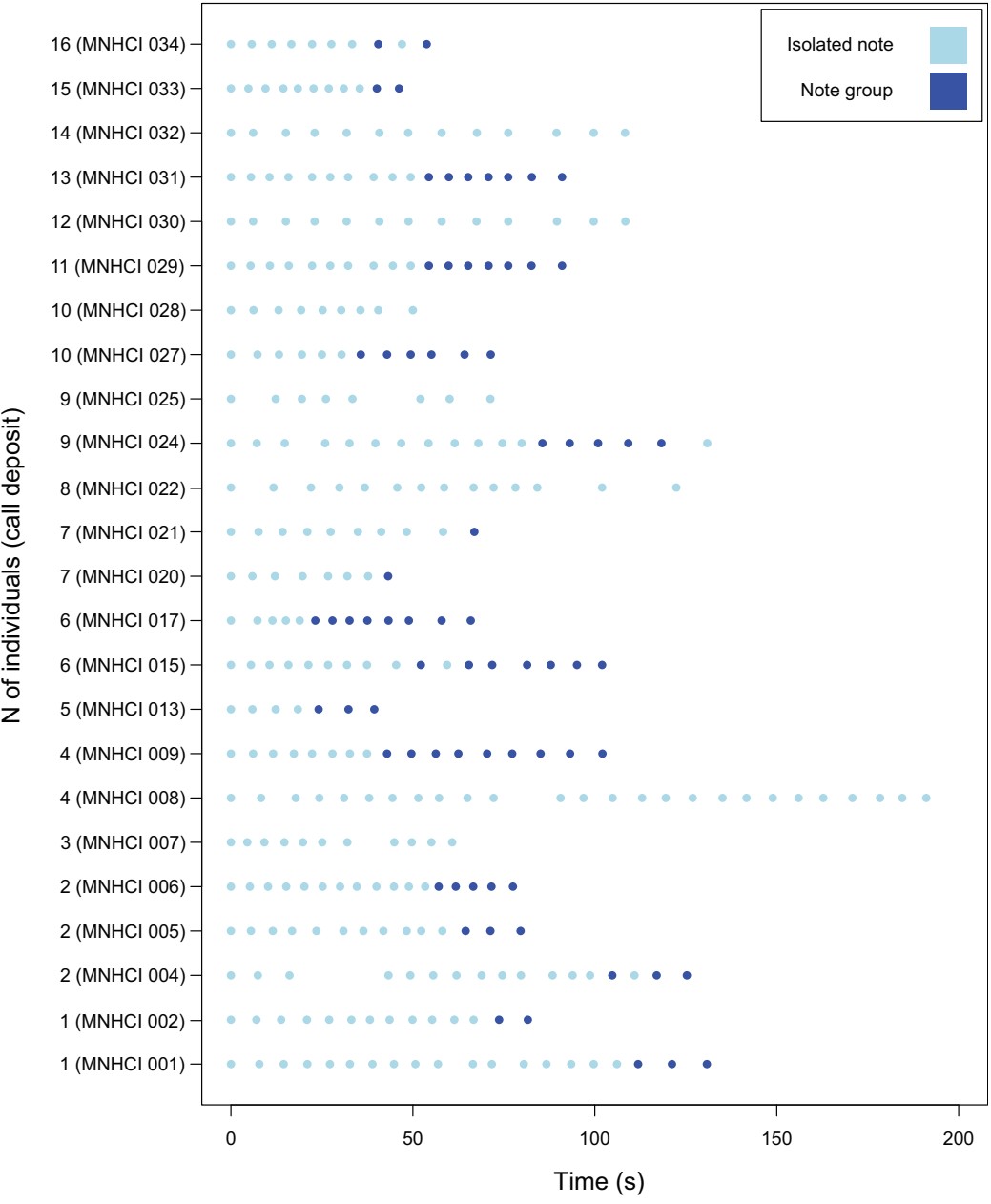

**Figure 2 Graphical representation of the emission of isolated notes and note groups of the advertisement calls (AC) of *B. albolineatus* (only AC recorded from the beginning were considered).** Note the individual variation. The number of pulses of each note can be observed in Table 2. Study site: MHNCI, Museu de História Natural Capão da Imbuia.

range of 0.41–60.32 s; see feature #3 in Fig. 1) as opposed to a mean of 53.71 s (±25.38 s; range of 18.39–98.90 s; Table 1; see feature #2 in Fig. 1) of the part of the advertisement calls with only isolated notes. The part of the advertisement call with only note groups also had a faster note rate, with 18.60 notes issued per minute, on average (±3.84 note per minute; range of 12.81–27.16 notes per minute; see feature #6

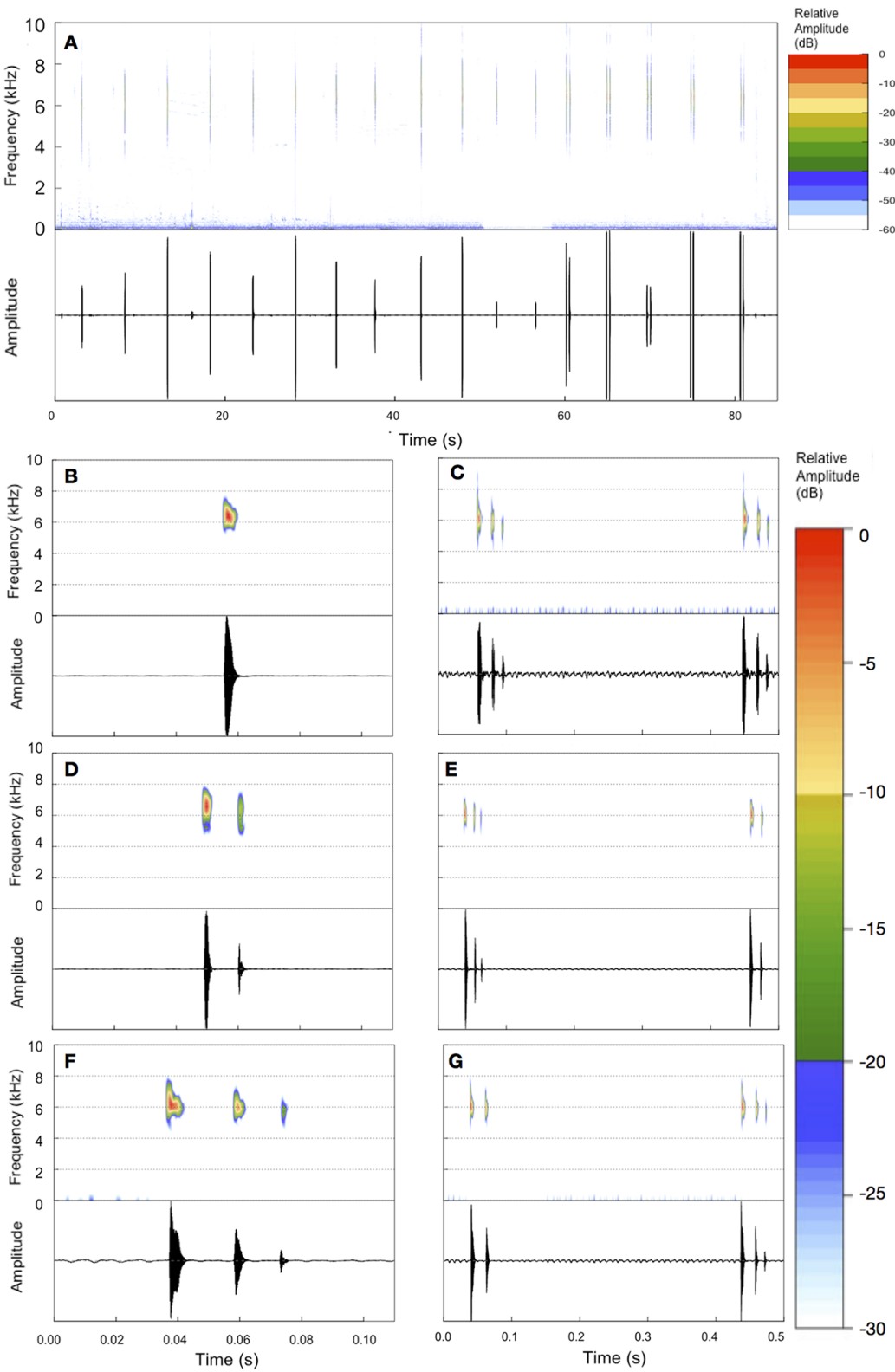

**Figure 3 Example of an entire advertisement call and also notes of other advertisement calls of *B. albolineatus*.** (A) Entire advertisement call (MHNCI 006; individual collected and housed at MHNCI 10296). (B, D, F) All examples observed of isolated notes, with one pulse (B = MHNCI 008), two pulses (D = MHNCI 022), and three pulses (F = MHNCI 026). (C, E, G) Examples of note groups, with 3–3 pulses (C = MHNCI 026), 3–2 pulses (E = MHNCI 027; individual collected and housed at MNRJ 90349), and 2–3 pulses (G = MHNCI 026). Study sites: MHNCI, Museu de História Natural Capão da Imbuia; MNRJ, Museu Nacional, Rio de Janeiro. Spectrograms produced with a FFT size of 4,096 points, Hann window, and overlap of 90% in (A) and FFT 128 points, Hann window, and overlap of 90% in (B–G).               

in Fig. 1), against 10.25 notes per minute on average (±1.61 note per minute; range of 7.28–13.62 notes per minute; Table 1) in the part of the call with isolated notes (when note groups occurs; see feature #5 in Fig. 1).

The number of pulses per isolated notes varies from one to three ($\bar{x} = 2.00 \pm 0.59$; Table 1; Fig. 3; see feature #10 in Fig. 1). The isolated notes that initiate the advertisement call do it with one pulse (eight advertisement calls) or two pulses (16 advertisement calls; Table 2). However, most of the isolated notes along the advertisement call escalated the number of pulses (1–2, 1–3 or 2–3; 18 advertisement calls), whereas the isolated notes maintained a constant number of pulses only in six of the advertisement calls (2–2; Table 2). The number of pulses per note in note groups varied from two to three ($\bar{x} = 2.70 \pm 0.46$; Table 1; Fig. 3; see feature #11 in Fig. 1). The total number of pulses in note groups varied from four to six ($\bar{x} = 5.40 \pm 0.82$; Table 1). Four combinations of number of pulses occurred in note groups (2–2 to 3–3), being more common the combination of 3–3 pulses (61.7%; Tables 1 and 2). All pulses both in isolated notes and note groups are interrupted units of the subsequent pulses, isolated by short moment of silence.

Note duration of isolated notes varies from 0.002 to 0.037 s ($\bar{x} = 0.020 \pm 0.007$s) and duration of note groups varies from 0.360 to 0.578 s ($\bar{x} = 0.465 \pm 0.053$ s; Table 1; see features #12 and #13 in Fig. 1). The inter-note interval in isolated notes is, on average, 6.663 s (4.092–12.248 ± 1.705 s; see feature #14 in Fig. 1) and the inter-note group interval is, on average, 6.871 s (4.322–10.678 ± 1.768 s; Table 1; see features #15 in Fig. 1). The inter-note interval within note groups is, on average, 0.412 s (0.319–0.526 ± 0.050 s; Table 1; see feature #16 in Fig. 1). The note dominant frequency varies from 5.34 to 7.32 kHz ($\bar{x} = 6.38 \pm 0.30$ kHz; Table 1). Finally, the highest frequency spans from 7.11 to 9.85 kHz ($\bar{x} = 8.44 \pm 0.49$ kHz) while the lowest frequency spans from 3.09 to 5.21 kHz ($\bar{x} = 4.07 \pm 0.45$ kHz; Table 1).

## DISCUSSION

In this study, we used a note-centered approach (*sensu Köhler et al., 2017*) to describe the advertisement call of *B. albolineatus*. We argue that there are two advantages for a note-centered approach to describe the calls of species of the *B. pernix* group. First, it is consistent with descriptions of calls of the species of the *B. ephippium* and *B. didactylus* groups (Table 3). For instance, in the *B. ephippium* group, the advertisement call of *B. crispus* has been described as "a long and low-intensity buzz with a regular repetition of notes" (*Condez et al., 2014*); the call of *B. darkside* "is characterized by pulsed notes emitted in extremely long sequences" (*Guimarães et al., 2017*), whereas the call of

**Table 2 Distribution of the number of pulses per note (separated by ",") along the advertisement calls (AC) of *B. albolineatus* (see features 10 and 11 in Fig. 1).**

| N of individuals (voucher number) | Number of pulses per note | Number of isolated notes we hear being emitted before recording the AC |
|---|---|---|
| 1 (MHNCI 001) | 1, 2, 2, 2, 3, 3, 3, 3, 3, 3, 3, 3, 3, 3, 3, 3, (3–3), (3–3), (3–3) | 0 |
| 1 (MHNCI 002) | 1, 1, 2, 2, 3, 2, 3, 3, 3, 3, 3, 3, (3–3), (3–3) | 0 |
| 2 (MHNCI 003) | 3, 3, 3, 3, 3, 3, 3, 3, 3, (3–3) | ? |
| 3 (MHNCI 004) | 2, 2, 2, 1, 1, 2, 2, 2, 3, 3, 2, 2, 3, (3–2), 3, (3–3), (2–2) | 0 |
| 3 (MHNCI 005) | 1, 2, 2, 2, 3, 3, 3, 3, 3, 3, 3, (2–2), (3–3), (3–3) | 0 |
| 3 (MHNCI 006) | 2, 2, 2, 2, 3, 3, 3, 3, 3, 3, 3, (3–3), (3–3), (3–3), (3–3), (3–3) | 0 |
| 4 (MHNCI 007) | 1, 1, 2, 2, 2, 2, 2, 3, 3, 3, 3 | 0 |
| 5 (MHNCI 008) | 1, 1, 1, 2, 2, 2, 2, 2, 2, 1, 1, 2, 2, 2, 2, 2, 2, 2, 2, 1, 2, 2, 2, 2 | 0 |
| 5 (MHNCI 009) | 2, 2, 3, 3, 3, 3, 3, 3, (3–3), (3–3), (3–3), (3–3), (3–3), (3–3), (3–3), (3–3), (2–3) | 0 |
| 5 (MHNCI 010) | 2, 3, 3, 3, 3, 3, 3, (3–3), (3–3), (3–3), (3–3), (3–3), (3–3) | ? |
| 6 (MHNCI 011) | 2, 1, 1, 2, 1, 2, 2, 1, 2, 1, 2, 2, 1 | 3 |
| 6 (MHNCI 012) | 2, 2, 2, 2, 2, (3–2) | 2 |
| 6 (MHNCI 013) | 2, 2, 2, 2, (2–2), (2–2), (2–2) | 0 |
| 7 (MHNCI 014) | 2, (3–2), (3–2), (2–2) | ? |
| 8 (MHNCI 015) | 2, 3, 3, 3, 3, 3, 3, 3, 2, (3–3), 2, (3–3), (3–3), (3–3), (3–2), (3–3), (3–2) | 0 |
| 8 (MHNCI 016) | 2, 2, 2, 2, 3, 3, 3, 3, 3, (3–3), 3, (3–3), (3–3), (3–3), (3–3), (3–3), (3–3), (3–2) | 3 |
| 8 (MHNCI 017) | 2, 2, 3, 2, 3, (3–2), (3–3), (3–3), (3–2), (3–3), (3–3), (3–3), (2–3) | 0 |
| 8 (MHNCI 018) | (3–2), (3–3), (3–3), (3–2), (3–3), (3–3), (3–3), (3–3) | ? |
| 9 (MHNCI 019) | 2, 2, 2, (2–2), 2, (2–2), (2–2), (2–2), (2–2) | ? |
| 9 (MHNCI 020) | 2, 2, 2, 2, 2, 2, 2, (2–2) | 0 |
| 9 (MHNCI 021) | 2, 2, 2, 2, 2, 2, 2, 3, (2–2) | 0 |
| 10 (MHNCI 022) | 2, 2, 2, 2, 3, 3, 3, 3, 3, 3, 3, 3, 2, 2 | 0 |
| 11 (MHNCI 023) | 2, 2, 2, 2, 2, 2 | ? |
| 12 (MHNCI 024) | 2, 2, 2, 2, 2, 2, 2, 2, 3, 2, 2, (3–3), (3–3), (2–3), (3–3), (3–3), 2 | 0 |
| 12 (MHNCI 025) | 2, 2, 3, 3, 3, 2, 3, 3 | 0 |
| 13 (MHNCI 026) | 2, 3, 3, (3–2), 3, (3–2), (3–3), (2–3), (3–3), (3–3), (3–3), (2–3) | ? |
| 14 (MHNCI 027) | 2, 2, 2, 3, 3, 3, (3–2), (3–3), (3–3), (3–3), (3–3), (3–3) | 0 |
| 14 (MHNCI 028) | 1, 2, 2, 2, 2, 3, 3, 3, 2 | 0 |
| 15 (MHNCI 029) | 2, 2, 2, 2, 2, 2, 2, 2, 2, (2–2), (2–2), (2–2), (2–2) | 0 |
| 16 (MHNCI 030) | 1, 1, 2, 2, 2, 2, 2, 2, 2, 2, 2, 2, 2, 2, 2, 2, (2–2), (2–2), (2–2), (2–2) | 0 |
| 17 (MHNCI 031) | 2, 2, 3, 3, 3, 3, 3, 3, 3, 3, (3–3), (3–3), (3–3), (3–3), (3–3), (3–3), (3–3) | 0 |
| 18 (MHNCI 032) | 2, 2, 2, 2, 2, 2, 2, 2, 2, 2, 2, 2, 2 | 0 |
| 19 (MHNCI 033) | 1, 1, 2, 1, 2, 2, 2, 2, (2–2), (2–2) | 0 |
| 20 (MHNCI 034) | 2, 2, 2, 2, 2, 2, 2, (2–2), 2, (2–2) | 0 |

**Note:**
Pulses per note groups are indicated between parenthesis, but indicating separately by "–" the number of pulses in each particular note of the group (see Figs. 1 and 3).

*B. pitanga* "[...] consists of low-intensity pulsed notes uttered in a long series" (*De Araújo et al. (2012)*; see *Pombal, Sazima & Haddad (1994)* for a similar description in the case of *B. ephippium*). Likewise, in the *B. didactylus* group, the call of *B. hermogenesi*
**Table 3 Comparison of the features used to describe the advertisement call of *Brachycephalus*.**

| Feature | B. pernix group | | B. ephippium group | | | | | | | B. didactylus group | |
|---|---|---|---|---|---|---|---|---|---|---|---|
| | B. albolineatus | B. tridactylus | B. crispus | B. darkside | B. ephippium | B. ephippium | B. pitanga | B. pitanga | B. pitanga | B. hermogenesi | B. sulfuratus |
| Call duration (s) | 39,933–191.141 (88.367 ± 35.733) [24/16] | ?–? (0.11 ± 0.02) [?/17][1] | ?–300 (? ± ?) [5/?] | 2.9–66.2 (30.4 ± 25.3) [7/5] | 120–360 (? ± ?) [?/?] | | | | | 0.2–1.9 (? ± ?) [?/?][2] | 1.5–2.3 (1.8 ± 0.2) [95/11] |
| Call rate (calls per second) | | | | | | | | | | ?–? (0.19 ± ?) [?/?][2] | |
| Interval between calls (s) | | | | 6.2, 11.2 [2/?] | | | | | | | 3.1–7.4 (5.1 ± 1.4) [95/11] |
| Note rate (notes per minute) | 5.891–18.088 (11.439 ± 3.216) [24/16] | | | 186.4–243.4 (211.4 ± 25.6) [5/?] | | | ?–? (159 ± 11) [?/2] | | | | |
| Note rate (notes per second) | | ?–? (0.16 ± 0.03) [11/?] | ?–? (1.67 ± 0.09) [5/?] | | | | | | | ?–? (1.09 ± ?) [?/?][2] | 0.1–0.3 (0.2 ± 0.0) [485/11] |
| Pulse rate (pulses per second) | | | ?–? (17.4 ± 2.12) [5/?] | 36.8–78.4 (56.9 ± 4.9) [790/5] | | | ?–? (62 ± 8) [?/2][3] | | | | 6.1–12.3 (9.3 ± 1.8) [?/11] |
| Number of notes per call | 8–29 (17.26 ± 6.38) [27/16] | 1 | | 9–253 (114 ± 97.1) [7/5] | | | | | | 1–7 (? ± ?) [?/?][2] | 4–7 (5.3 ± 0.9) [485/11] |
| Number of pulses per isolated notes | 1–3 (2.00 ± 0.595) [323/20] | 0 | 7–12 (10 ± 1.19) [100/5] | 5–8 (6.3 ± 0.7) [790/5] | 5–15 (12 ± 1.96) [57/?] | | ?–? (11.1 ± 1.2) [?/2] | 6.90–14.30 (10.86 ± 1.62) [?/?] | | | 7–11 (8.8 ± 1.3) [?/11] |
| Number of pulses per note in note groups | 2–3 (2.70 ± 0.459) [230/16] | | | | | | | | | | |
| Note duration of isolated notes (s) | 0.002–0.037 (0.020 ± 0.007) [96/19] | ?–? (0.11 ± 0.02) [?/17] | ?–? (0.28 ± 0.02) [100/5] | 0.083–0.163 (0.111 ± 0.014) [790/5] | 0.093–0.125 (0.112 ± 0.006) [19/?] | | ?–? (0.170 ± 0.013) [?/2] | 0.15–0.25 (0.19 ± 0.03) [400/40] | | | 0.131–0.233 (0.195 ± 0.013) [485/11] |
| Duration of note groups (s) | 0.360–0.578 (0.465 ± 0.053) [62/16] | | | | | | | | | | |
| Pulse duration (s) | | | ?–? (0.027 ± 0.004) [517/5] | | | | | | | | 0.02–0.03 (0.024 ± 0.005) [?/11] |

(Continued)

| Feature | B. pernix group | | B. ephippium group | | | | | | | B. didactylus group | |
|---|---|---|---|---|---|---|---|---|---|---|---|
| | B. albolineatus | B. tridactylus | B. crispus | B. darkside | B. ephippium | B. ephippium | B. pitanga | B. pitanga | B. pitanga | B. hermogenesi | B. sulfuratus |
| Inter-note interval in isolated notes (s) | 4.092–12.248 (6.663 ± 1.705) [62/15] | | ?–? (0.35 ± 0.02) [100/5] | 0.122–0.215 (0.159 ± 0.014) [783/5] | 0.123–0.149 (0.134 ± 0.007) [18/?] | | | 0.20–0.43 (0.28 ± 0.05) [400/40] | | | |
| Inter-note group interval (s) (15) | 4.322–10.678 (6.871 ± 1.768) [32/13] | | | | | | | | | | |
| Inter-note interval within note groups (s) | 0.319–0.526 (0.412 ± 0.050) [55/16] | | | | | | | | | | |
| Note dominant frequency (kHz) | 5.340–7.321 (6.376 ± 0.304) [256/10] | ?–? (4.8 ± 0.2) [?/17] | ?–? (4.6 ± 0.19) [100/5] | 2.856–3.797 (3.382 ± 0.185) [790/5][4] | | ?–? (3.94 ± 0.24) [?/5] | ?–? (4.9 ± 0.2) [?/2] | 4.311–5.550 (4.816 ± 0.414) [400/40] | ?–? (5.43 ± 0.30) [?/8] | | 6.2–7.2 (6.7 ± 0.3) [?/11] |
| Call dominant frequency (kHz) | | | | | | | | | | ?–? (6.8 ± 0.8) [5/?][5] | |
| Highest frequency (kHz) | 7.113–9.852 (8.437 ± 0.492) [145/19] | 6.4 (? ± ?) [?/17][6] | ?–? (5.7 ± 0.17) [100/5][6] | | 5.3 (? ± ?) [?/?][6] | | | | | | 8.2–10.3 (9.3 ± 0.3) [?/11][6] |
| Lowest frequency (kHz) | 3.092–5.212 (4.066 ± 0.448) [145/19] | 3.2 (? ± ?) [?/17][6] | ?–? (3.5 ± 0.19) [100/5][6] | | 3.4 (? ± ?) [?/?][6] | | | | | | 4.5–5.5 (4.9 ± 0.3) [?/11][6] |
| 5–95% frequency[7] | | | | 2.484–5.766 (? ± ?) [?/?] | | | | | | | |
| "Highest sound pressure" (dB) | | ?–? (110 ± 5.6) [?/17] | | | | ?–? (47.0 ± 5.7) [3/?] | 56–66 (? ± ?) [4/?] | | ?–? (57.6 ± 1.8) [8/?] | | |
| Approach (sensu Köhler et al. (2017)) | Note-centered | Call-centered | Note-centered | Note-centered | Note-centered | Not applicable | Note-centered[2] | Note-centered | Not applicable | Note-centered[2] | Note-centered |
| Source | This study | Garey et al. (2012) | Condez et al. (2014) | Guimarães et al. (2017) | Pombal, Sazima & Haddad (1994) | Goutte et al. (2017) | De Araújo et al. (2012) | Tandel et al. (2014) | Goutte et al. (2017) | Verdade et al. (2008) | Condez et al. (2016) |

**Notes:**
Values are expressed by: range (mean ± SD) [sample/individuals].
[1] Represents note duration under note-centered approach.
[2] Note-centered approach and call-centered approach probably mixed in this measurement.
[3] The unit of measure was erroneously cited as Hz.
[4] Feature cited as "peak frequency" by *Guimarães et al. (2017)* but refers to our dominant frequency.
[5] We are not sure if in the measurement was not mixed with note dominant frequency.
[6] The measurement procedure has not been explained and data may be not comparable.
[7] Feature cited as "dominant frequency" by *Guimarães et al. (2017)*.

"may be simple, constituted by a single note, or complex, composed of groups of two to seven similar notes" (*Verdade et al., 2008*), whereas the call of *B. sulfuratus* is "long, composed of a set of four to seven high-frequency notes [...] repeated regularly" (*Condez et al., 2016*). In all those cases, the call was considered as the entire sequence of notes. On the other hand, *Garey et al. (2012)* considered single notes as calls and largely overlooked any patterns over longer periods of time. Second, using a note-centered approach facilitates comparisons with calls of congeners, as well as underscores the considerable differences in call structure between species in a single group and among species groups.

There are only a few species of *Brachycephalus* with described advertisement calls. In Table 3, we summarize all data and features used in those descriptions. It is striking the extent to which descriptions vary in the number of features used and in how often they lacked important details, such as methodological procedures and sample size. These issues make it difficult to conduct a more precise comparison with the call of *B. albolineatus*. Nevertheless, *B. albolineatus* is the only known species with an advertisement call that is structurally modified along its emission, i.e., more structured (with notes with increasingly more pulses and with note groups). However, as stated above, we do not rule out the possibility that the advertisement call of *B. tridactylus* indeed exhibits some level of structuring such as that of *B. albolineatus*. Another striking difference is how much the note of *B. albolineatus* is shorter than that of *B. tridactylus* (*Garey et al., 2012*), both of the *B. pernix* group (average of 0.020 and 0.11 s, respectively). *B. albolineatus* have a very reduced number of pulses in isolated notes in comparison with the species of the *B. ephippium* and *B. didactylus* groups, i.e., a mean of two pulses against means of 6.3 pulses in *B. darkside* (*Guimarães et al., 2017*), 10.0 pulses in *B. crispus* (*Condez et al., 2014*), 10.9 and 11.1 pulses in *B. pitanga* (*De Araújo et al., 2012*; *Tandel et al., 2014*), and 12 pulses in *B. ephippium* (*Pombal, Sazima & Haddad, 1994*; Table 3), in species of the *B. ephippium* group, and against a mean of 8.8 pulses in *B. sulfuratus*, of the *B. didactylus* group (*Condez et al., 2016*; Table 3). *B. albolineatus* has the highest interval in the range of note dominant frequency, that include a variation of two kHz, only slightly comparable to the range variation of 1.2 kHz of *B. pitanga* (*Tandel et al., 2014*; Table 3). Meanwhile it is premature to provide a discussion about this variation, given that most of the available data of dominant frequency in *Brachycephalus* only report their average values (Table 3). It should be noted that the large frequency range of the "dominant frequency" for *B. darkside* presented by *Guimarães et al. (2017)*, including a variation of 3.3 kHz, is not comparable to the variation in *B. albolineatus* because the measurement refers to a frequency range (Table 3). The one-pulse notes of *B. albolineatus* may represent "warming notes" (*sensu Bornschein et al., 2007*), which refers to notes beginning a call and that are attenuated (e.g., less intense (less audible)), although one-pulse notes also appear along the call in some advertisement calls.

In relation to the number of pulses, the homology between what has been termed as a note and a pulse in *Brachycephalus* is still uncertain (Table 3). One way of standardizing this criterion would be to associate a note with an airflow, that is,

with an increase and decrease of the vocal sac (*McLister, Stevens & Bogart, 1995*). This would be an objective form of definition and standardization, but it would require one to overcome the difficulty in observing a *Brachycephalus* calling, or even better, to associate the movement of the vocal sac with the spectrogram. A possible result of this analysis could be verifying that the pulses could be defined as notes in some species.

Apparently, there is a trend of individuals of *B. albolineatus* to invest progressively more energy along the emission of each particular advertisement call. There are three sources of evidence for this: (1) advertisement calls normally escalated, incorporating note groups at the last third part of the call (76%); (2) pulses per note increased during the emission of isolated notes (up to three; 62%); and (3) note groups usually had 3–3 pulses per note (62%), which is the combination of the groups with highest number of pulses (Table 2). Intra-individual variation analysis also demonstrated that less structured advertisement calls (i.e., with notes with less pulses) are not fixed individually and can vary in the course of an hour. In the only species of the *B. pernix* group with its advertisement calls described to date, *B. tridactylus* (*Garey et al., 2012*), there was no evidence of escalation in structure. It is possible that the advertisement calls with isolated notes and note groups could have distinct functions, perhaps territorial defense when composed only by the former and territorial defense plus mating when composed by isolated notes and note groups. There is a parallel between the differences of isolates notes vs note groups of *B. albolineatus* and the "territorial call"/"aggressive call" vs advertisement call of *B. pitanga* (*De Araújo et al., 2012*) and *B. darkside* (*Guimarães et al., 2017*). In both of these territorial/aggressive calls there are shorter notes with reduced number of pulses that in the advertisement calls, like the isolated notes of *B. albolineatus* that span 0.002–0.037 s ($\bar{x} = 0.020$ s) and have one to three pulses ($\bar{x} = 2.0$ pulses) whereas note groups span 0.360–0.578 s ($\bar{x} = 0.465$ s) and have four to six pulses ($\bar{x} = 5.4$ pulses).

In a recent study, *Goutte et al. (2017)* suggested that *B. ephippium* and *B. pitanga* are insensitive to the sound of their own calls. This raises some questions about the validity of discussions about the possible reproductive and behavioral use of calls in the case of *B. albolineatus*, as well as for the use of calls in the taxonomy of the group. *Goutte et al. (2017)* suggest that calls may have been maintained in the studied species because of the call side effects (e.g., vocal sac movement) or by evolutionary inertia, for example. The relevant issue to be discussed here is that *B. ephippium* and *B. pitanga*, both members of the *B. ephippium* group, present vocal and visual behavior (vocal sac movements) above the leaf litter (*Goutte et al., 2017*), unlike *B. albolineatus* and all other species of the *B. pernix* group (Marcos R. Bornschein et al., 1988–2018, personal observation), which call exclusively under the leaf litter and vocal sac movements are not visible. We do not abandon the hypothesis that species of the *B. pernix* and *B. didactylus* groups have a more complete auditory system than *B. ephippium* and *B. pitanga* and the ability to perceive their own calls. This is an interesting subject brought only now to the fore and open to further discussion.

## CONCLUSIONS

*B. albolineatus* is the first species in the genus whose advertisement call has been recognized as increasing in complexity over the course of its emission. Its advertisement call is long and composed by isolated notes and note groups, which tend to be emitted during the last third of the call. Intra-individual variation demonstrates that calls can be composed only by isolated notes or by isolated notes and note groups in a subsequent call. The number of pulses per notes escalates along the call. These results underscore how a note-centered approach is able to reveal important aspects of the temporal dynamics of the advertisement call of the studied species.

## ACKNOWLEDGEMENTS

Helena Zarantonieli provided valuable administrative support for the projects conducted by Mater Natura - Instituto de Estudos Ambientais. Larissa Teixeira assisted in the preparation of Fig. 1. Lucas Forti and Carlos Araújo made valuable comments that greatly improved our work.

### Funding

Marcio R. Pie was supported through a grant from CNPq/MCT (301636/2016–8). Fieldwork was funded by Fundação Grupo Boticário de Proteção à Natureza, through a project (A0010_2014) conducted by Mater Natura - Instituto de Estudos Ambientais. The funders had no role in study design, data collection and analysis, decision to publish, or preparation of the manuscript.

### Grant Disclosures

The following grant information was disclosed by the authors:
CNPq/MCT: 301636/2016–8.
Fundação Grupo Boticário de Proteção à Natureza: A0010_2014.

### Competing Interests

Marcio R. Pie is an Academic Editor for PeerJ.

### Author Contributions

- Marcos R. Bornschein conceived and designed the experiments, performed the experiments, analyzed the data, contributed reagents/materials/analysis tools, prepared figures and/or tables, authored or reviewed drafts of the paper, approved the final draft.
- Luiz Fernando Ribeiro performed the experiments, contributed reagents/materials/analysis tools, authored or reviewed drafts of the paper.
- Mario M. Rollo Jr. analyzed the data, contributed reagents/materials/analysis tools, prepared figures and/or tables, authored or reviewed drafts of the paper.

- André E. Confetti contributed reagents/materials/analysis tools, authored or reviewed drafts of the paper.
- Marcio R. Pie contributed reagents/materials/analysis tools, authored or reviewed drafts of the paper.

## Field Study Permissions

The following information was supplied relating to field study approvals (i.e., approving body and any reference numbers):

We collected vouchers according to permits issued by ICMBIO–SISBIO (no. 20416–2).

## Data Availability

The original call recordings are provided as Supplemental Files.

## Supplemental Information

Supplemental information for this article can be found online at http://dx.doi.org/10.7717/peerj.5273#supplemental-information.

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
