# Peer review of "Advertisement call of Brachycephalus albolineatus (Anura: Brachycephalidae)"

_PeerJ, doi:10.7717/peerj.5273_

## Round 0.1 · original submission · Major Revisions

Both reviewers see merit in your manuscript but would like to see major revisions before it could be considered for publication.

While I do not have a problem with the length of introduction, I do agree with Carlos that you should not build the readers’ expectations, but limit your introduction to pertinent information on the data that you providing. Your paper is based on the calls of a single species, thus extended introduction and discussion on evolution of calls in the group appears unwarranted. There are other difficulties because your manuscript only addresses calls of a single species. It is hard for the reader to interpret the variability within a species without it being compared to another using your same methodology. The great variation in dominant frequency shown in Figure 4 isn’t displayed in any of the calls in Figure 3. Does this mean that the variation happens temporally? Do calls increase in frequency with time (i.e. warming/escalating)? And/Or is the variation a factor of the number of calls that you recorded per individual (some recordings are very short)? Lucas makes the point that the variation is unusually high (nearly 2kHz), and I feel that this is not adequately explained/explored methodologically, or even mentioned in the discussion. This is further muddied by some variables being omitted due to quality of recordings; Lucas provides comprehensive advice here.

The audio files uploaded sound very good. Could you include the specimen numbers so that these can be matched to the data provided in the ms? Alternatively, a spreadsheet which explains the relationship between the call numbers and the specimens. Could this be “N of individuals” or “call deposit” – not at all clear. Note that “N of individuals” usually stands for Number of individuals, but presumably not here.

·

Basic reporting

This manuscript present a description of the advertisement call of a microendemic Neotropical frog, which is poorly know. The knowledge about acoustic properties of different species are especially important for anurans taxonomy and behavior, and, then, papers like this can help to improve our biodiversity understanding. I am very excited to see good manuscripts with call descriptions coming to be published in PeerJ. This group of tiny frogs from Brachycephalidae family have many species with calls unknown, which raises the importance of this manuscript. The authors present a manuscript with a legitime intentionality in a well-written text. However, some issues, basically related to the acoustic and data analysis, were listed and should be improved upon before acceptance.

Experimental design

The sample effort is satisfactory and matches with the objective of the paper.

Validity of the findings

Questionable, considering the issues pointed specifically in the general comments.

Additional comments

Although the results are compelling, the note-centered approach used by the authors for recognize different advertisement calls of the same male is questionable, because it is not clear how the authors make the distinction of a call unit inside a sequence of calls. Other detail, more problematic, is that the concept of uninterrupted units of sound are not well interpreted for pulses analysis, once that, using this context, pulses need to be fused and not separated by space as the authors were using. In this case, they are counting notes, and not pulses. Considering this, the temporal analysis of advertisement calls is ruined, and should be reformulated or rethought. One of the main issues is that spectral data were, basically, neglected by the authors. Range, minimum and maximum frequency were not included, because the authors found a strong dependence of this data with the quality of recordings. So, why didn't they use only the good recordings for obtaining these values?
Other special issue is that dominant frequency presented an uncommon high variability. It could be real, but I understand that pre-editions and adequate use of a sample value in Fast Fourier Transform (FFT) for spectral measurements would alleviate this variation, and potentially may generate reliable measurements. Finally, the way that the variation on call properties were presented is out of the regular pattern used for describing acoustic data in such matter. Call properties are regularly classified as static or dynamic, depending on the coefficient of variation calculated by: (standard deviation / average) multiplied by 100 (see: Gerhardt HC. 1991. Female mate choice in treefrogs: static and dynamic acoustic criteria. Animal Behaviour 42(4):615–635). There is a well-structured discussion in the literature about why some call traits present high variability, while others are very stereotyped, mainly concerning to social context, species recognition and temperature effects. See the discussion in Forti, L.R. et al. 2017: The advertisement call of Sphaenorhynchus caramaschii. Further, the variation in the species level may help to establish boundaries among sister species. To the moment, we do not know if species from the pernix group are insensitive to calls, and, while someone do not prove something different, calls still are sexual signals used for intraspecific communication, and they are useful for species diagnosis. Such subjects are totally missing in the manuscript.
Some specific issues are: The authors said that “We made numbered markings close to the recorded individuals” for avoiding the recordings of the same males. However, how were made this “markings” for such tiny frogs? How was this controlled? Were the distance among neighbors measured? Their territory size was previously calculated? Were studied the males movement?
In the conclusion the authors said that some call traits may be related to individual arousal levels. And, they have concluded that, the variation on calls composition are possibly related to defense of territory and female preference. However, they do not have evidences to link these relationships, and once these thoughts are only speculations, it should be out of the conclusion. All these issues have to be addressed in the manuscript.

·

Basic reporting

The paper describes the advertisement call the recently described Brachycephalus albolineatus. The paper is well written, and relevant data is shown.
I found the introduction to be quite long, way beyond what would be expected for a call description paper. For example, it discusses in detail the evolutionary history of the genus, and the differences among its three phenetic groups (lines 51-93). Even though it is interesting, it induces the reader to expect a comparative analysis based on call parameters, which would allow for the investigation of the evolutionary history of the group. As put by the authors themselves on line 104: “Given that Brachycephalus is a group with mostly allopatric species, it is of great interest to investigate the evolution pattern of their calls”. However, no formal comparative analysis was made, and the discussion comparing the described call to those of other species is not compelling, and lacks of focus. The authors also mention that: “Over the course of our studies on mountain frogs from the southern Atlantic Rainforest, in which we discovered several new species of Brachycephalus (Pie & Ribeiro 2015, Ribeiro et al. 2015, Bornschein et al. 2016b, Ribeiro et al. 2017), we recorded the advertisement calls of nearly all of them”, but here they described the advertisement call of a single species. The authors are clearly missing the chance of making the evolutionary investigation (what is seems to be proposed in the introduction), while describing the advertisement calls of several Brachycephalus species in the process. As I see, there are two distinct directions for the paper. Either the authors simplify the manuscript by making a call description paper with a straightforward introduction, results and discussion; or really investigate call evolution within the group, describing the calls they do have, and making a comparative analysis that might include distances comparisons (phylogenetical, acoustical or geographical), or examining the presence phylogenetic signals in species’ calls.

I really hope the suggestions made here does not disappoint the authors, while helping them in preparing a new improved version of the manuscript for submission.

Experimental design

The study aims to investigate the evolution pattern of the calls of the genus Brachycephalus, but fails to do so on its present format. Nevertheless, it elegantly describes the advisement calls of Brachycephalus albolineatus. The way I see, there are two possible roadmaps to be followed. The paper can either be simplified, aiming to describe the advertisement call of the species, or it could formally include a comparative analysis, which could formally investigate the call evolution patterns of the group.
I feel most of the tables should be moved to supplementary materials, for they fail to support the paper’s discussion and objectives, even though the data could provide helpful information for upcoming work. A single table could give full support for a call description paper, by describing the mean (or median) of each parameter (e.g. dominant frequency, pulse rate, note rate etc.) of the call. Additionally, I feel figure 1 could be improved by the use of a measurement map, which shows, by using a sonogram/oscilograma of a call, the measurements made. I really don’t feel the scheme of fig 1. has worked. I had a hard time trying to understand what was represented there. I also don’t see the necessity of figures #2 and #4. If the authors wish to show the macrostructure of the call, they could do this in figure 3, by showing a spectrogram of a call with a longer time window. For instance, spectrogram D works well to show the pulse number variation showed in the first column (A, C, E), while an even longer spectrogram (maybe with 30s or more) could show the intended macrostructure in figure 2. Either way, the spectrograms are great. As for figure 4, I really don’t see its purpose, what is the intention in showing individual variations in dominant frequencies? These differences are present in most calls (the coefficient of variations are usually between 10-20%), and unless the authors have a clear objective for the inclusion of figure 4, I would advise to remove it.

Validity of the findings

The paper should either be reformulated to a call description paper, or to a comparative analysis. In its current form, it seems to propose a comparative analysis, which was not made.

Additional comments

l.61-63: Since this is a paper on acoustical communication, it might be important to cite that the process of miniaturization might also affect species hearing abilities (SILVA, H. R., CAMPOS, L.A., SEBBEN, A. 2007. The auditory region of Brachycephalus and its bearing on the monophyly of the genus (Anura: Brachycephalide). Zootaxa 14122: 59-68.)
l.66-93: Unless the paper really tackles the acoustical evolution of the group, I feel these two paragraphs are beyond the objectives of the paper, and should be completely removed.
l. 104 Even though the authors recognize the importance of call descriptions, and sustains that the lack of descriptions are surprising due to the conspicuous acoustic behavior of the group (l.97), they only describe the call of a single species. This is surprising, specially as the authors clearly announce that they have recordings of at least 10 species of the genus (l.108-111)!
l.119-220: There is no need to mention discarded recordings.
l.121-124: There is no need for SISBIO licenses for sound recordings, unless these are made within federal parks. Not really sure why the information on these licenses are necessary.
l.124: Please consider starting the methods by describing the study area. From the text I am not sure whether the recordings were made within the type locality.
l.132: It seems some of the spectrograms were build with edited files, in which the intensity was normalized to 0dB before the analysis. Please clear out what the editing consisted of.
l.137: Even though noise filters could affect the signal, a high pass filter at 2kHz, for example, could reduce low frequency noise such as depicted in spectrograms B, E & F (Figure 3). Please consider the use of high pass (sometimes band pass) filters followed by normalization as a standard editing procedure. This procedure allows for direct comparisons between the analyzed calls. Please (http://dx.doi.org/10.1016/j.anbehav.2012.04.026)
l.143: I am not sure what the authors considers to be a “conspicuous spectrogram”, and have no idea what they mean by “compare to the oscilogams”. Compare how? Please elucidate.
l.145: If a note is composed by pulses, it cannot be composed of “uninterrupted units of sound”, for silence should be found between the pulses that forms a note.
l.178: It is not clear to me what the authors named isolated notes. It does not seem to be a single note, but rather notes issued with a much lower emission rate (even though the inter note interval seems to be around 6s for both; l. 221). If this is the case, and considering that such differences in the emission rate could code for distinct messages, it seems to me there are two repertoire components being described here, not a single one. Both components are named as advertisement calls by the authors, but discrete acoustical differences such as displayed here are usually associated with two messages, not a single one. For instance, differences on emission rates and number of pulses are the main alterations found between territorial and advertisement calls of B. pitanga (Araújo et al. 2012, cited by the authors). Did the authors considered the possibility of two repertoire components? Either way, please better describe what are you calling isolated notes.
l.202: Calls from different individuals are rarely so stereotyped, and a similar variation are found elsewhere. For example, Verdade et al. 2008 (cited by the authors) describes a standard deviation of about 1kHz within B. hermogenesi dominant frequencies. Hence, unless the authors wish to test the use of these calls to transfer information on individuals (what does not to seems to be the case), I suggest removing this portion of the ms.
l.240: Even though we agree that the absence of standardization prevents a detailed comparison, there are call parameters that are present in most call descriptions of the group and could be uses in such comparisons, such as dominant frequency, number of pulses, pulse rate, note rate, note duration and inter-note interval. Additionally, as the authors already possess recordings of nearly 10 species, the description of those calls could allow for species comparisons.
l.242: Most call descriptions of Brachycephalus were made based on “call-centered approach”, and it seems the authors have not measured the entire call, as defined here (e.g. Verdade et al. 2008, Araújo et al. 2012). The analysis of a portion of the advertisement call (as defined here) has been widely used for species with long calls, such as Rhinella (http://dx.doi.org/10.11646/zootaxa.3784.1.9), even though the description of the entire call have also been made for the genus (http://dx.doi.org/10.2994/057.005.0209). It could be that the calls are too long, making it difficulty to record from the start. Nevertheless, it is possible to make comparisons with the note parameters described here (even though it would be considered a call in other studies).
l.256: the authors have not measured call intensity.
l.258: Even thought Sandra Gautte’s paper is quite interesting, I would take it cautiously. There is substantial morphological modifications in the hearing apparatus of the group (Zootaxa 1422: 59–68 (2007)), probably related to its minituriarization (Biological Journal of the Linnean Society 99: 752-767 (2010)). The authors did not examine in full the possibility that m. opercularis would have a central role in Brachycephalus hearing abilities, not bony elements. Additionally, due to the presence of multiple repertoire components (e.g. Araújo 2012), it would be quite counter intuitive for these frogs to have lost their hearing abilities. In sum, even though the quality of the work done by Sandra Goutte and its collaborators, I would take it cautiously, and wait for further papers with the group before assuming the frogs do not hear.

---

## Round 0.2 · Minor Revisions

Thanks for your revision, which I agree is close to being ready for publication. The first reviewer has raised a series of objections to your response to the reviewers comments.

On the first point: "There is a suggestion for classifying individual notes by a single cycle of airflow back and forth from the lungs to the vocal sacs (see Gerhardt, 1998)"

I don’t think that you are going to reach a consensus with this reviewer. For this reason, I suggest you pick up this thread in the discussion including the reference to Gerhardt (as I presume you do not have this data).

The other points are useful, and I think that you should consider all of them carefully. The reviewer has looked at your calls and makes a number of useful suggestions.

Lastly, I agree that you should avoid terms like "we believe" and ask you to rephrase the same in the abstract and discussion (L49 & L207). You could instead use "We consider" or simply remove the statement of belief.

In addition, I have the following comments:
L32: Change to style of PeerJ for citing papers in the abstract, or remove citation
L45: calls as is proceeds - correct grammar
L48: could have distinct function - correct grammar
L57 & L77- update to Frost 2018?

·

Basic reporting

In general, the work have advanced in quality, the inclusion of a table comparing acoustic properties among species is very informative and useful as a “short review” about the current knowledge of bioacoustics in the genus. I congratulate the authors by the detailed review and improvements. However, some appointments in the first review have not been solved, even being necessary as showed in this new review. I really want to see this paper published as soon as possible, because it is a very important contribution (handled with a clear determination), but, first, some details should be adjusted before a final acceptance.
Historically, terms as pulses, notes, calls, chirps already have been used to label the same kind of basic acoustic units. Even after reviews of the subject as Toledo et al. (2014) and Köhler et al. (2017), still there are authors with particular interpretations. Such arbitrariness is always subject of confusion, and I understand that the authors defined call units by the note-centered approach described in Köhler et al. 2017, and that is ok. However, there is an unclear conceptualization yet. The authors are so emphatic that comparisons are limited due to the lack of a standard analysis procedure, but how can someone reproduce the measurements if there is no precision to define how long should be the silence interval among two different calls? Calls periodicity should be well described, once further works may test their functions.
I agree with the authors that pulses can be separated by silence interval (which is clear in “Köhler” et al. 2017), but the criteria adopted for notes as “uninterrupted sound units” was contradictory, once that using this definition and seeing the figure 3, each note would be an isolated pulse, because they are not fused. However, listening the calls present in the supplement files, I could better understand the authors’ view, once the note (using their definition) sounds very quick, even considering discrete intervals of silence among pulses. The interval of silence among pulses is noticeable only in the waveform (not by human hearing). The decision of take off the detail “uninterrupted sound units” to avoid confusion is a reasonable option. Although, there is a suggestion for classifying individual notes by a single cycle of airflow back and forth from the lungs to the vocal sacs (see Gerhardt, 1998). I don´t know if authors have saw males calling (because they apparently emit calls under the leaf litter), but this definition for notes, in this case, sounds very comfortable, and may avoid confusion. Besides, nothing changes in the results.
The authors wrote: “maximum and minimum frequency, are highly dependent on the recording conditions” - which conditions are those? For justifying the absence of these variables in the first version of the manuscript the authors argue that other papers have not included it. This is a naive and scientifically poor argument. If other authors have failed in such respect, they gonna fail too. This journal and society expect the best of knowledge, not just the acceptable.
Seems that the authors have included these properties in the manuscript without understand how they could be measured, once they declare "there is no clear limit of the beginning and ending of minimum and maximum frequency". I must say, all acoustic signals, in all spectral gradient, has a maximum and minimum frequency very accessible if they are not masked by background noise. Professional bioacoustics software, as Raven, measures it with precision. In Raven you should use a tool in the “choose measurements” called “Frequency 5% (Hz)” and “Frequency 95% (Hz)”. Once you selected your call units, the values will appear in a table. This tool measures the minimum and maximum frequency based in the energy distribution along spectral domain, considering a confidence interval of 5% inside the selection. Peak normalization process, which I suggested in the first review (not carried), it is an important step for standardization of amplitude among notes, which may avoid biases related to the variation on intensity, making call units more comparable and this tools (Freq 5% and 95%) more reliable. The method used by the authors (made by eyes, directly inspecting a colorful spectrogram) is very inaccurate, as they themselves assumed. I suggest the way I commented above.
About the variability in dominant frequency, the authors argue “We could expect this variability, since it reflects individual variation, but we do not understand how a pre-edition and an adequate use of a specific value in FFT size could change this result and yield more reliable measurements.” And they follows with a comparison of DF based in rough graphical interpretations with different FFTs. If the authors know the significance of FFT, they must understand that spectrograms generated by high FFT sizes should increase precision in spectral measurements, because the energy will be sampled inside smaller spectral classes (frames), see the figure 1 in the zenodo repository for understanding (one note of B. albolineatus represented by two FFTs).
Figure 1 in: https://doi.org/10.5281/zenodo.1193816
Then, it is clear that more accurate spectral measurements were achieved with FFT of 1024. When we transpose it to the right way of obtaining these variables in Raven, some differences could be noticed. Then, I repeated the test applied by the authors:
Test 1 – FFT = 126: DF = 5857 Hz, MaxFre =7235 Hz , and MinFreq = 5512 Hz
Figure 2 in: https://doi.org/10.5281/zenodo.1193816
Test 2 – FFT = 256: DF = 5857 Hz, MaxFre =7062 Hz , and MinFreq = 5512 Hz
Figure 3 in: https://doi.org/10.5281/zenodo.1193816

Test 3 – FFT = 512: DF = 5857 Hz, MaxFre =6976 Hz , and MinFreq = 5512 Hz
Figure 4 in: https://doi.org/10.5281/zenodo.1193816

Test 4 – FFT = 1024: DF = 6847 Hz, MaxFre =7019 Hz , and MinFreq = 5598 Hz
Figure 5 in: https://doi.org/10.5281/zenodo.1193816

In all examples, we find spectral differences among the FFTs tested.
The authors declared: “it is possible to see differences in the resolution of spectrograms” – Of course we might like to have fine temporal and spectral resolution in a spectrogram, although these two demands are almost incompatible. However, adjustments in window overlap enable considerable resolution improvements without lost of data.
See the spectrograms in the figure 6 (Zenodo repository below) comparing the same call generated by FFT of 1024, but with different window overlap.
Figure 6 in: https://doi.org/10.5281/zenodo.1193816
Clearly a simple adjust in the window overlap value recovered the image resolution, and values of frequencies are exactly the same.
In theory, lost of resolution with a larger FFT window is only on the time scale and this can have consequences in visual inspections if the overlap is not increased (usually it is fixed at 50% in Raven). I usually do the temporal measurements using the waveform, and spectral analysis with readable spectral composition in possible higher FFTs, explored by settings in the "window overlap". Then, as the authors can see by the figures above, a higher FFT should have been used. I hope, this time, the authors would interpret my suggestion with good eyes, that usually lower FFTs can affect spectral measurements, because they are less precise.
The authors refused the suggestion of a proper variation analysis, saying that they do not have a large number of recordings for each individual. This is a weak argument. Many acoustic properties could be explored in their material, since there are properties that not involving variations among calls, as spectral variation among notes, notes duration and pulses by notes.
In the comment 6, the authors manifested: “There is no guarantee that calls are always useful for species diagnosis” as other justification for the absence of a proper variation analysis. The sentence cited by Köhler et al. 2017 is justified mainly because body size can affect frequencies, however, sexual signals are generally good evidences for diagnosis (not guarantee), and should help to raise interesting questions about taxonomy. Further works may take advantage of such evidences to test if different populations, with different calls, could be different species, for example.
I have a particular final worry about the same subject, because the authors used a superficial view about acoustic variation declaring: “we simply believe in individual variation and not in environmental or population interferences.” First of all, in science there are no “beliefs”, there are facts, evidences and suggestions. Second, such sentence is inconsistent with ecological trends as the acoustic adaptation hypothesis or the Lombard effect, which have already been verified in previous works (see: Shen & Xu, 2016; and Goutte et al., 2016). I recommend the reading of Wilkins et al. (2012) - Evolutionary divergence in acoustic signals: causes and consequences. This is a complete review about how acoustic signals are possibly modulated.

Experimental design

Acoustic analysis have some issues to solve.

Validity of the findings

No comment

Additional comments

Some very punctual suggestions were made directly in the text, and, in my view, the manuscript presented a good advance, which included important corrections raised by the other reviewer too.

·

Basic reporting

I have carefully read the revised version of the manuscript "Advertisement call of Brachycephalus albolineatus (Anura: Brachycephalidae)", and I am extremely happy with the work done by the authors. They have considered most of the suggestions made by both referees, and as a result the paper have improved much. I have no further comments, and as I see, the paper is now ready for publication.

Experimental design

no further comments.

Validity of the findings

no further comments.

Additional comments

I was very happy to see this second version of the paper, and have no further comments on the manuscript.

---

## Round 0.3 · accepted · Accept

Thanks for your revision, which I find acceptable for publication in PeerJ. There is one minor detail that needs correction. Line 253 states: "...to capture its call in a slow-motion video...", but slow-motion video does not capture sound, so this needs to be corrected to something that indicates synchronized recording of sound and video.

#